# FEDERATED LEARNING FOR INFERENCE AT ANYTIME AND ANYWHERE

## ABSTRACT

Federated learning has been predominantly concerned with collaborative training of deep networks from scratch, and especially the many challenges that arise, such as communication cost, robustness to heterogeneous data, and support for diverse device capabilities. However, there is no unified framework that addresses all these problems together. This paper studies the challenges and opportunities of exploiting pre-trained Transformer models in FL. In particular, we propose to efficiently adapt such pre-trained models by injecting a novel attention-based adapter module at each transformer block that both modulates the forward pass and makes an early prediction. Training only the lightweight adapter by FL leads to fast and communication-efficient learning even in the presence of heterogeneous data and devices. Extensive experiments on standard FL benchmarks, including CIFAR-100, FEMNIST and SpeechCommandsv2 demonstrate that this simple framework provides fast and accurate FL while supporting heterogenous device capabilities, efficient personalization, and scalable-cost anytime inference. Our anonymous code for reviewing can be found here.

## 1 INTRODUCTION

Federated learning (FL) was proposed by McMahan et al. (2017) as a new paradigm for distributed learning in which user data privacy is protected. Following the introduction of the FL setting, subsequent work focused on addressing the emerging challenges that arise due to FL constraints, such as communication cost Mishchenko et al. (2022), data heterogeneity Li et al. (2020) and supporting diverse device hardware Horvath et al. (2021); Rapp et al. (2022). For example, to reduce the communication cost, ideas borrowed from model compression, such as quantization Alistarh et al. (2017); Fu et al. (2020), sparsification Stich et al. (2018) and pruning Yu et al. (2021); Jiang et al. (2022) have been successfully applied; to mitigate the non-IID issue of data heterogeneity, different model training recipes for optimization Li et al. (2020); Wang et al. (2020b), model initialization Nguyen et al. (2022) and architecture design Qu et al. (2022) have also been proposed.

A new question has now emerged for FL community: Can we benefit from the recent success of large-scale centralized pre-training of foundation models Bommasani et al. (2021)? Although contemporary federated learning has predominantly been concerned with collaborative training of deep models from scratch McMahan et al. (2017); Li et al. (2020), neglecting publicly available pre-trained models, it has been observed by Qu et al. (2022) that fine-tuning pretrained vision transformers (ViT) significantly improves FL performance for various image recognition tasks and enables great robustness to the data heterogeneity among clients. Despite being an important step forward, fine-tuning the whole pre-trained ViT can be problematic due to the heavy communication cost of exchanging large numbers of model parameters and the weak capabilities of on-device training for many client devices. In this paper, we address this problem by reframing FL as a parameter-efficient (PE) downstream learning task. This is in line with the recent PE-based adaptation developments in centralized vision and natural language processing methods. This line of parameter-efficient adaptation research includes adapters Rebuffi et al. (2017); Houlsby et al. (2019); Tomanek et al. (2021), prompt tuning Li and Liang (2021); Lester et al. (2021), bias-only fine-tuning Zaken et al. (2021) and so on. We contribute a new adaptor suited for FL under the foundation model regime, which is designed for the requirements of adaptation to fit client devices at *anytime* (under different compute and memory budgets) and *anywhere* (under severe data heterogeneity).

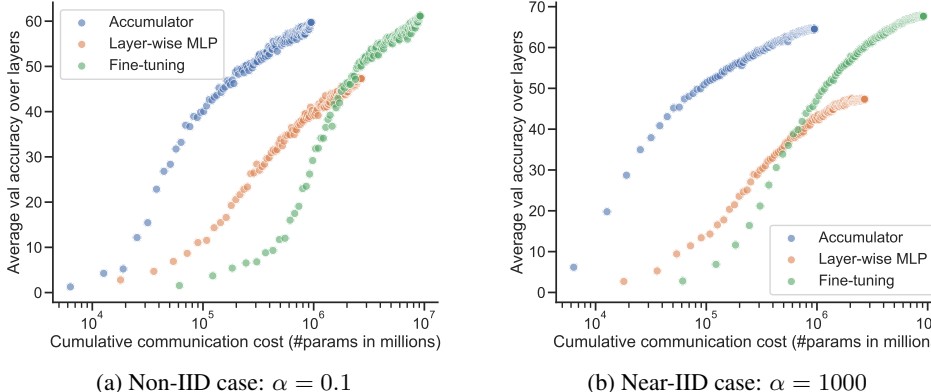

(a) Non-IID case: $\alpha = 0.1$          (b) Near-IID case: $\alpha = 1000$

Figure 1: The effectiveness of Accumulator-based federated FM adaptation under anytime and anywhere setting in terms of communication cost and classification performance. The experiments are conducted on CIFAR-100. Full details can be found in Section 5. Each point corresponds to an evaluation during FL, where the cumulative communication cost measures the communication of gradients w.r.t. model parameters between clients and the server. We would like to emphasize that our Accumulator a) converges faster (less communication cost) than the baselines regardless of the data heterogenity condition and b) performs as well as the upper bound – full-model fine-tuning.

Given a pre-trained Transformer model, e.g. ViT Dosovitskiy et al. (2020), DeiT Touvron et al. (2021) and AST Gong et al. (2021), we re-wire its feature extraction pathway to tackle the anytime and anywhere challenges. Specifically, we keep track of the `CLS` tokens after each self-attention transformation and make use of the history of previous `CLS` tokens to revise the current `CLS` token by a lightweight Transformer, which is termed *Accumulator*. Our Accumulator has an order of magnitude fewer parameters than a pre-trained Transformer model and is the only module trainable during the local forward and backward propagations; therefore, the training and communication efficiencies can be both significantly improved. To show this, we make a comparison between Accumulator and a standard early-exit (Laskaridis et al., 2020) model (*Layer-wise MLP*, by inserting for each self-attention block a MLP classification head) and the full-model fine-tuning (Qu et al., 2022; Nguyen et al., 2022). The comparisons for non-IID, and IID cases are presented in Figure 1, which clearly show that our method is more efficient in reaching a certain target performance in both cases. In addition, due to the efficient optimization enabled by our Accumulator, user personalization for a particular client can be conducted efficiently with even better performance than fine-tuning the whole model.

The contributions of our work are the follows:

- We take a different perspective to the existing FL literature and propose a parameter-efficient learning method to adapt the pre-trained Transformer FMs in FL scenarios.

- We propose a novel parameter-efficient adapter, which modulates all layers of a pre-trained Transformer FM and allows flexible early predictions (anytime inference).

- Extensive experiments on standard FL benchmarks, including CIFAR-100, FEMNIST and SpeechCommandsv2, show our method can improve global accuracy, personalization and communication efficiency with excellent robustness to data and compute heterogeneities.

## 2 RELATED WORK

### 2.1 FEDERATED LEARNING

Though federated learning is still an emerging topic, there are already many works published in the literature. There exist two general FL settings, *cross-device* McMahan et al. (2017) and *cross-silo* Heikkilä et al. (2020) FL. In this paper, we focus on the former. The main research focuses in this setting are designing systems to solve communication efficiency, data and system heterogeneity problems. Researchers have proposed different techniques to improve communication efficiency.

E.g. Alistarh et al. (2017); Fu et al. (2020) proposed to use the quantized model parameters or gradients during the FL communication. Similarly, Stich et al. (2018) sparsified, and Yu et al. (2021); Jiang et al. (2022) pruned the training model into smaller cardinality to reduce communication cost. And some researchers tried to address this by progressive model training Wang et al. (2022). Another main focus is on data heterogeneity. In contrast to centralized training where the learner has access to the whole distribution, each worker having access to a biased distribution in non-i.i.d. FL negatively affects convergence and final model accuracy. In attempts to alleviate this, people proposed to add proximal regularization in the local training termed FedProx Li et al. (2020). Alternatively, some normalized averaging technique to mitigate the inconsistency between different clients is proposed in Wang et al. (2020b). Interestingly, more recently, researchers found that model initialization (pre-trained v.s. random) plays an important role in reducing the detrimental impact of heterogenienty Nguyen et al. (2022), and so does model architecture (Transfomer v.s. CNN) Qu et al. (2022). System heterogeneity is also a concern in cross-device FL, where different participants may have different hardware resources and thus be unable to perform the same amount of learning. Some researchers used nested-dropout Rippel et al. (2014) as a nicely fit method to address the varied of computational constraints between different clients Horvath et al. (2021), as a different dimension from considering early-exit networks Laskaridis et al. (2020).

Our work differs from the existing literature. Rather than training from scratch, we focus on the challenges and opportunities of adapting a pre-trained Transfomer FM by FL. While this might seem to exacerbate communication bottlenecks and system heterogeneity issues above, we show that with our novel adapter module we can simultaneously ameliorate all the above challenges of communication cost, heterogeneous device capabilities, and difficulty of federated learning on non-i.i.d. data. Our work goes substantially beyond Qu et al. (2022) and (Nguyen et al., 2022), who just focus on conventional federated fine-tuning (which we treat as a baseline), to support communication efficient federated fine-tuning with support for heterogeneous device capabilities – thanks to our novel adaptation module.

## 2.2 PARAMETER-EFFICIENT LEARNING

The idea of parameter-efficient learning of adapters was first proposed in Rebuffi et al. (2017) for adapting a single model into multiple datasets. It has since been extended into various problems, including few-shot learning Li et al. (2022) and ASR Tomanek et al. (2021). Especially as the pre-trained large-scale FMs' model sizes are soaring significantly, parameter-efficient (transfer) learning has become important in NLP. Instead of fine-tuning the full pre-trained model, people lean toward designing different small set modules for adapting the pre-trained FMs into downstream tasks Li and Liang (2021); Lester et al. (2021); Houlsby et al. (2019). Li and Liang (2021) found that tuning the prompt input of the pre-trained language model enables excellent performance on downstream tasks. While Houlsby et al. (2019) found that fine-tuning some injected adapters can be more effective than fine-tuning the top layers of a pre-trained NLP model. In this work, we make the first attempt at parameter-efficient adaptation in an FL context, developing a novel transformer-based adaptation module specifically customized for this task.

## 3 PRELIMINARIES

### 3.1 FEDERATED LEARNING

Let us consider a typical setting of FL with $K$ devices, where a local device $i$ has $N_i$ private training examples denoted by $\{(\mathbf{x}_j, y_j)\}_{j=1}^{N_i}$ with $\mathbf{x}_j$ the input image and $y_j$ the target label. The learning objective following McMahan et al. (2017) aims at finding a model parameter $w$ that minimizes the weighted average loss over all local devices:

$$w = \arg\min_w \sum_{i=1}^{K} \alpha_i \mathcal{L}_i(w), \quad \mathcal{L}_i(w) = \frac{1}{N_i} \sum_{j=1}^{N_i} \ell(F_w(\mathbf{x}_j), y_j), \tag{1}$$

where $\alpha_i = N_i / \sum_{i=1}^{K} N_i$, $\ell(F_w(\mathbf{x}_j), y_j)$ is the task-specific loss function and $F_w()$ is the formed model. The main difference that renders the problem difficult is we can only compute $\mathcal{L}_i(w)$ on device $i$ to protect the privacy for the user. The common setup (see more details in Beutel et al.

(2020)) introduces a server to receive gradients sent from each client device, and therefore brings two major challenges: communication cost and data heterogeneity.

**Personalization.** Minimizing Eq. 1 explicitly optimizes the generalization of the shared global model $w^*$ across all clients. However, the model performance on individual clients might still be sub-optimal, especially under client data heterogeneity. Clients often care more about personalized performance (i.e., overfitting to client data). Therefore, given the outcome of global federated model learning in Eq. 1 denoted by $w^*$, each client $i$ can then further fine-tune the parameters locally to obtain personalized parameter $w_i$

$$w_i = \arg\min_{w^*} \frac{1}{N_i} \sum_{j=1}^{N_i} \ell(F_{w^*}(\mathbf{x}_j), y_j), \tag{2}$$

### 3.2 PARAMETER-EFFICIENT LEARNING

Parameter-efficient learning is a typical strategy for adapting pre-trained FMs to downstream tasks. The full model size of an FM is often much larger than the size of downstream task data making the fine-tuning prone to overfitting. Not mentioning the back-propagation over the full FM is extremely expensive. To this end, multiple PE learning methods Houlsby et al. (2019); Rebuffi et al. (2017); Tomanek et al. (2021); Li and Liang (2021); Lester et al. (2021); Zaken et al. (2021) have been proposed for fast adaptation of FMs, in which the learning objective is typically formulated as

$$w_{\text{PE}} = \arg\min_{w_{\text{PE}}} \frac{1}{M} \sum_{j=1}^{M} \ell(F_{w_{\text{FM}}, w_{\text{PE}}}(\mathbf{x}_j), y_j), \tag{3}$$

where $w_{\text{FM}}$ corresponds to the frozen pre-trained foundation model, $w_{\text{PE}}$ is the weight associated to the introduced parameter-efficient module and $\{(\mathbf{x}, y)\}^M$ are the $M$ data pairs from the downstream task of interest. Our goal in this paper is to design such a lightweight module for Transformer-based FMs in the context of FL.

## 4 ADAPTING TRANSFORMER FMS INTO FEDERATED LEARNING

The motivation of our work is to treat federated learning as a downstream task for adapting a pre-trained Transformer FM. In the following sections, we will introduce the two main modules, including a pre-trained Transformer and our attention-based adapter Accumulator, and how Accumulator modulates the outputs of a Transformer to support the early predictions. The overview of our model architecture is depicted in Figure 2.

### 4.1 TRANSFORMER MODEL

A Transformer model typically consists of a sequence of residual blocks of multi-head self-attention (MSA), each followed by a residual block of feed-forward multilayer perceptron (MLP) with Layer-Norm (LN) applied to both MSA and MLP blocks. Denote by $\mathbf{x}$ the input, $\mathbf{p}$ the positional encoding, and $\mathbf{z}^l := [z_{\text{cls}}^l, z_1^l, \ldots, z_N^l]$ the intermediate tokens, the feed-forward pass of a Transformer can be formalized as

$$\mathbf{z}^0 = \text{Tokenizer}(\mathbf{x}) + \mathbf{p}, \tag{4}$$

$$\mathbf{z}^l = \text{MSA}(\text{LN}(\mathbf{z}^{l-1})) + \mathbf{z}^{l-1}, \quad l = 1 \cdots L, \tag{5}$$

$$\mathbf{z}^l = \text{MLP}(\text{LN}(\mathbf{z}^l)) + \mathbf{z}^l, \quad l = 1 \cdots L. \tag{6}$$

### 4.2 ACCUMULATOR

To adapt a pre-trained Transformer model, we inject our Accumulator into each[1] self-attention block followed by a shared MLP head to enable early predictions. Formally, by collecting the history of

---

[1]Note that there is only one Accumulator that handles the outputs from all layers.

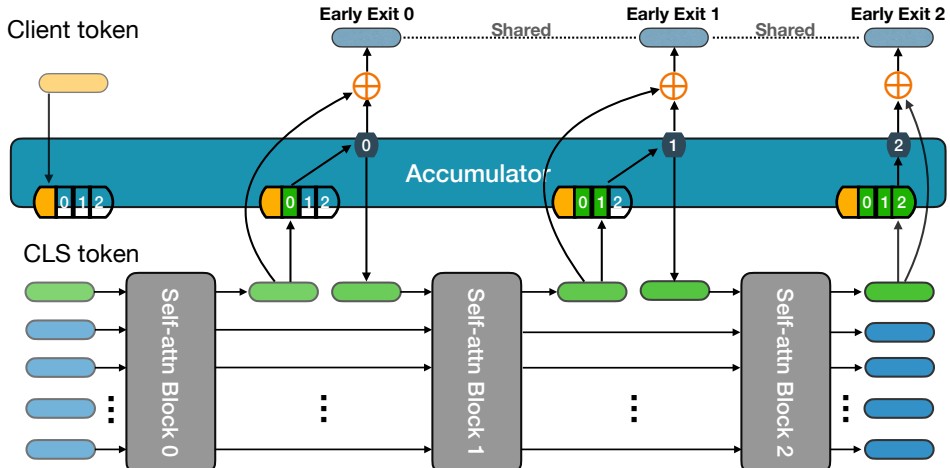

Figure 2: Model architecture of the proposed Accumulator. Given a frozen Transformer FM (in grey), the Accumulator aggregates the information from the history of the CLS tokens through the transformation, which yields a better feature representation for early exits at different layers. The early exit slots are numbered (there are three exits in this example), and the corresponding pathways are denoted by different line types (e.g., dotted for exit 0 and dashed for exit 1). A residual connection is introduced to each early exit between the CLS token at the exit and the output of the Accumulator. A special CLS token (or referred to as Client token) is introduced to be the first token of the Accumulator, which can be tuned for personalization while freezing the other parts.

CLS tokens, we replace $z_{\text{cls}}^l$ by

$$z_{\text{cls}}^l \leftarrow h_{l+1}^l \quad \text{with} \quad h^l = \text{Accumulator}_\phi([z_{\text{client}}, z_{\text{cls}}^0 + p_0', \dots, z_{\text{cls}}^l + p_l']), \tag{7}$$

where the Accumulator parameterized by $\phi$ is another Transformer (randomly initialized) with a single[2] self-attention block as defined by Eq. 5 and 6. $h_{l+1}^l$ means the $l+1$ element in $h^l$. In our Accumulator, the positional embedding $\mathbf{p}' := [p_0', \dots, p_L']$ embeds the layer information from a pre-trained Transformer. Note that the modified $z_{\text{cls}}^l$ will be again used by the pre-trained Transformer, specifically, the next self-attention block, to produce $\mathbf{z}^{l+1}$.

For early predictions, instead of taking the modified $z_{\text{cls}}^l$, we find it is better to make use of the client token and the original $z_{\text{cls}}^l$, specifically, the task-specific prediction at exit $l$ is given by

$$\hat{y}^l = \text{MLP-head}_\psi(h_0^l + z_{\text{cls}}^l), \tag{8}$$

where MLP-head parameterized by $\psi$ is the early exits shared across all layers.

### 4.3 FEDERATED LEARNING OF ACCUMULATOR

Following the PE optimization in Eq 3, the learnable parameters $w_{\text{PE}} = (\phi, \psi)$ reduces to the weights of the Accumulator and the MLP head.

**Learning.** Now the learning problem in Eq. 3 becomes

$$\arg\min_{w_{\text{PE}}} \sum_{i=1}^K \frac{\alpha_i}{M} \sum_{j=1}^M \ell(\hat{y}_j^{l_i}, y_j), \tag{9}$$

where $l_i$ is the layer of an early exit for client $i$, which is set for each inference in client $i$. Assuming the client $i$ has the budget using up to $L_i$ layers, there are two schemes for choosing the value of $l_i$: a) $l_i = L_i$ and b) $l_i$ is taken uniformly at random from the range $\{1, \dots, L_i\}$. We attribute this property of our formulation as *anytime* to accommodate clients with different computing and memory capacities.

---

[2]We find in Table 5 that increases from a single block to 3 blocks yields little improvement in accuracy but a significantly heavier communication burden.

**Personalization.**     There are two choices: fine-tuning the whole Accumulator $(\phi, \psi)$ or the `Client` token $z_{\text{client}}$ only. Given that they are both lightweight, the personalization is less likely to overfit even under an extremely low data regime.

**Inference.**     Given an FL trained model in client $i$, it can infer labels $\hat{y}^{L_i}$ not only from block $L_i$ according to its capability but also labels $\hat{y}^{l_i}$ as long as $l_i$ is smaller than the budget constraint $L_i$. This property can be very useful in the case when a cellphone suffers from low battery, which enables the user to set a small budget $l_i < L_i$ to allow energy-efficient inference.

## 5    EXPERIMENTS

### 5.1    EXPERIMENTAL SETUP

**Dataset Settings and Implementation.**     To verify the efficacy of our proposed FL framework, we conduct experiments on the standard FL benchmarks with Flower codebase Beutel et al. (2020), including CIFAR-100 Krizhevsky et al. (2009), FEMNIST Cohen et al. (2017) and Speech-CommandV2 Warden (2018), as downstream tasks of pretrained DeiT Touvron et al. (2021) and AST Gong et al. (2021). We use the original test set in CIFAR-100 as the global test dataset and split 10,000 images from the original training set as the personal validation set for each client when conducting the personalization experiments on CIFAR-100-C. We simulate three different data partitions for CIFAR-100, including one IID-data partition ($\alpha = 1000.0$), and two non-IID data partitions($\alpha = 1.0, 0.1$) by LDA partitioning following prior works Karimireddy et al. (2020); Wang et al. (2020a) with 100 clients. FEMNIST has a total of 80,000 images (grayscale) of 62 different character classes (10 numeric, 26 lowercase, and 26 uppercase). For FEMNIST, we partition the training data in two ways, one IID and one non-IID, as in Cohen et al. (2017) with 381 clients. The Speech Commands v2 dataset consists of 105,829, 16KHz 1-second long audio clips of a spoken word. And we conduct our experiment with the 12-classes version, with 10 classes as "yes", "no", "up", "down", "left", "right", "on", "off", "stop", "go" and one class "unknown" and a class "silence" which has no spoken word in the clip. The dataset has three disjoint sets of speakers: 2112 for training, 256 for validation and 250 for the test.

For each round in FL, we sample $10\%$ of clients for training on CIFAR100 and FEMNIST, and $1\%$ on SpeechCmdv2. We use a pretrained DEiT-small Touvron et al. (2021) model with $16 \times 16$ patches on ILSVRC-2012 Fei-Fei et al. (2010) as the foundation model, which can be used for both image and speech recognition – i.e. the backbone of AST Gong et al. (2021).

All experiments are conducted with Pytorch on a single Nvidia Tesla V100 GPU with results repeated three times and reporting the mean and std. All models are trained using an SGD optimizer with a cosine annealing learning rate schedule. The detailed recipe can be found in the Appendix A.

**Early Exits.**     Each client has its own computing capability. According to their capability, they can support a certain amount of FLOPs for one inference, or one FL round. For example, given a pretrained Transformer, some clients may be able to pass the input data through all layers of the Transformer to train a classifier. Some could only pass through half of them to the same end. Specifically, how many layers of a pre-trained transformer can be passed through during the local model training for a client depends on the client's computing capability. One can train a classifier head to make predictions for each layer to support the different compute capabilities among clients. These layer-wise classifier heads are often called early exits Laskaridis et al. (2021); Leontiadis et al. (2021). For example, we can straightforwardly train 12 early exit classifiers for a given pretrained DeiT-small, which has 12 layers. And then, those exits can be used for different clients accordingly. Briefly, we can treat 12 exits corresponding to 12-tiers of client compute capability – namely, the different levels of computing budgets in terms of anytime inference.

**Baselines.**     We run all our FL experiments with the FedAvg McMahan et al. (2017) strategy. We compare with all the baselines, namely, *i*) *Fine-tuning* Qu et al. (2022), where the whole pretrained vision transformer is finetuned end-to-end in FL; *ii*) *Layer-wise Linear* (L.W. Linear), where we append a learnable linear head after each transformer block[3] to be learned by FL, while keeping the rest frozen; *iii*) *Layer-wise MLP* (L.W. MLP), where we append a two-layer MLP with GELU and Random Dropout after each transformer block, with the rest frozen. *iv*) Our *Accumulator*, where

---

[3]We also tried with shared linear head among layers, but it achieved worse results.

Table 1: Conventional FL performance.

| Method | CIFAR-100 | | | FEMNIST | | SpeechCmdV2 |
|---|---|---|---|---|---|---|
| | IID (1000.0) | Non-IID (1.0) | Non-IID (0.1) | IID | Non-IID | |
| Fine-tuning | 85.05±0.55 | 84.91±0.98 | 84.25±1.14 | 86.87±0.74 | 85.28±1.45 | 98.22 |
| Linear head | 74.37±0.42 | 73.85±0.74 | 72.79±0.83 | 74.02±1.33 | 72.35±0.52 | 69.47 |
| + PA | 83.28±0.67 | 82.57±1.12 | 81.41±1.28 | 82.34±1.56 | 81.60±0.66 | 94.74 |
| MLP head | 76.49±0.43 | 75.69±0.81 | 74.44±0.95 | 74.26±1.69 | 72.87±0.69 | 74.33 |
| + PA | 84.55±0.69 | 83.87±1.25 | 82.33±1.24 | 82.49±1.88 | 81.54±0.78 | **95.63** |
| Accumulator | 84.90±0.75 | 84.34±1.33 | 83.31±1.54 | 79.12±1.64 | 78.05±0.75 | 93.24 |
| + PA | **85.35**±0.49 | **85.11**±1.28 | **84.02**±1.32 | **83.68**±1.75 | **82.24**±0.84 | **95.27** |

Table 2: Anytime FL performance.

| Method | CIFAR100 | | | FEMNIST | | SpeechCmdV2 |
|---|---|---|---|---|---|---|
| | IID (1000.0) | Non-IID (1.0) | Non-IID (0.1) | IID | Non-IID | |
| Fine-tuning | 67.24±1.13 | 66.95±1.93 | 60.05±2.51 | 76.43±1.45 | 75.82±2.42 | 93.71 |
| L.W. Linear | 36.42±0.80 | 36.06±1.35 | 34.00±1.77 | 35.92±1.23 | 35.49±1.74 | 65.10 |
| + PA | 47.33±0.95 | 47.94±1.53 | 47.11±1.92 | 57.36±1.45 | 56.78±1.89 | 84.16 |
| L.W. MLP | 37.62±0.76 | 37.84±0.98 | 38.22±1.34 | 35.27±1.29 | 34.54±1.78 | 64.57 |
| + PA | 48.65±0.89 | 48.39±1.02 | 47.94±1.21 | 55.21±1.48 | 54.42±2.01 | 83.19 |
| Accumulator | 64.28±1.01 | 63.02±1.79 | 57.34±2.23 | 75.47±1.41 | 75.12±1.86 | 84.66 |
| + PA | **65.23**±1.26 | **64.28**±1.84 | **58.40**±1.67 | **76.55**±1.63 | **76.03**±2.13 | **88.30** |

we append one shared Accumulator and one shared MLP into each transformer block, with the rest frozen. We also compare the variants of L.W. Linear, MLP, and Accumulator with Parallel Adapters (PA), as proposed in He et al. (2021), injected into the feed-forward MLP networks of the pretrained Transformer as their efficient and effective adaptation He et al. (2021).

**Evaluation Settings.** We evaluate all the methods in four different settings, including (1) *Conventional FL* – Clients train the local models by using only the final exit of a pretrained transformer, and the trained global model will be evaluated at a test set using only the final exit. (2) *Anytime FL* – Clients train the local models by using a random exit at each iteration, and the trained global model will be tested at each exit. (3) *multi-tier FL* – Clients train the local models by using a specific early exit determined by the tier of each client. The trained global model will be tested at each exit, and (4) Personalization, where the FL-trained model from setting (3) will be further finetuned using the local data in each client.

## 5.2 EXPERIMENTAL RESULTS

**Conventional Federated Learning.** Results in Table 1 show the comparison between all competitors. The results show that fine-tuning works the best among all methods. This is unsurprising, as it has access to the large combined dataset of all clients and can use this to tune the whole model, but it incurs the most local training and communication costs. Among the other competitors, we can see that our Accumulator achieves the best results already when used alone. Parallel adapters are quite effective in enabling federated adaptation of the pretrained Transfomer, boosting the performance of base methods L.W. Linear and MLP significantly in all settings, especially on SpeechCmdv2. Nevertheless, our Accumulator is complementary with parallel adapters, achieving the best result in all cases, with some results even surpassing the Fine-tuning method, such as on CIFAR-100 with IID and Non-IID (1.0). The results demonstrate the efficacy of our proposed Accumulator to adapt a pretrained Transformer model into FL at anywhere under any type of data heterogeneity.

**Anytime Federated Learning.** In this setting, we will consider the early exit situations. For each layer of a pretrained DeiT, we train an early exit, such as a linear head, an MLP head and our Accumulator. At each local training iteration, an exit layer index $l \in [0, \cdots, 11]$ will be sampled randomly, and only that exit (and accumulator where relevant) will be trained. For fine-tuning, we append the layer-wise MLP heads and train all parameters during the FL training. After training, the global model will be evaluated on the test set at all exits. The results in Table 2 and Figure 3

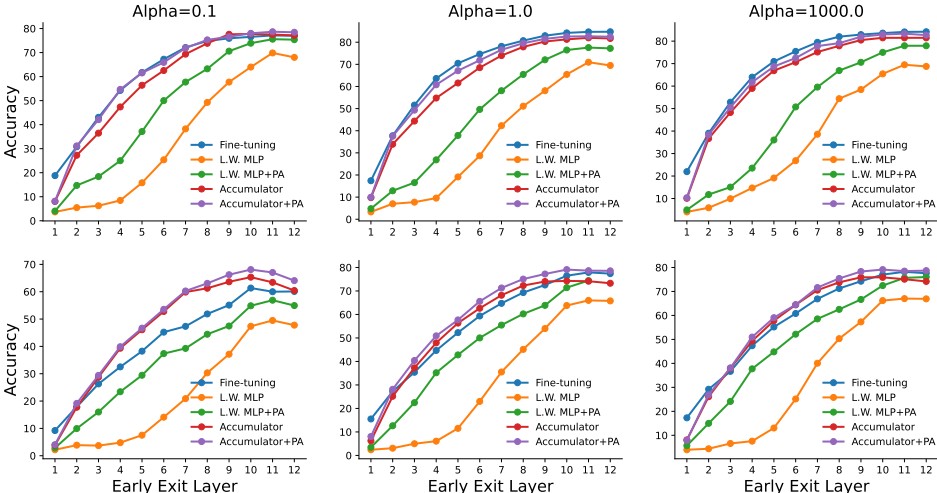

Figure 3: Test accuracy of each exit trained on CIFAR-100 with anytime federated learning (Top) and multi-tier constraint (Bottom). The memory and FLOPs cost of each exit is given in Table 7.

Table 3: Multi-tier based FL performance.

| Method | CIFAR100 | | | FEMNIST | | SpeechCmdV2 |
|---|---|---|---|---|---|---|
| | IID (1000.0) | Non-IID (1.0) | Non-IID (0.1) | IID | Non-IID | |
| Fine-tuning | 58.17±0.52 | 56.70±0.83 | 42.47±1.41 | 74.79±0.75 | 74.36±1.36 | 93.03 |
| L.W. Linear | 32.43±0.34 | 30.98±0.60 | 21.33±0.97 | 35.68±0.54 | 35.17±1.01 | 65.14 |
| + PA | 46.96±0.41 | 45.14±0.68 | 33.94±1.03 | 57.06±0.60 | 54.67±1.15 | 83.23 |
| L.W. MLP | 36.57±0.32 | 34.38±0.58 | 24.37±0.93 | 33.12±0.59 | 32.88±1.13 | 64.84 |
| + PA | 51.33±0.47 | 49.29±0.66 | 36.29±1.07 | 51.88±0.67 | 51.39±1.27 | 82.28 |
| Accumulator | 58.27±0.56 | 57.32±0.88 | 47.26±1.18 | 72.14±0.72 | 71.33±1.39 | 85.21 |
| + PA | **59.33**±0.72 | **58.48**±0.96 | **48.55**±1.30 | **72.67**±0.81 | **72.05**±1.57 | **87.42** |

*top* show the average and budget-wise performance over all exits. Without surprise, we can see that global fine-tuning achieves the best at substantial comms cost. Among the more efficient competitors, we can see that our Accumulator outperforms L.W. Linear and MLP significantly when no extra adapters are used. With parallel adapters, all methods enjoy a performance boost, leading our Accumulator to the best performance in all cases, outperforming fine-tuning in a few situations.

**Multi-tier based Federated Learning.** More realistically, there is a certain level of system heterogeneity among client devices. Individual devices in FL training have a certain level of computing capability and their associated early exit should be persistently fixed in both training and testing. Results in the setting are evaluated and reported in Table 3, where we can see that most results dropped to some extent compared with Table 2. This is expected due to the existence of system heterogeneity among clients. And most observations in Table 3 are similar to the previous tables. One interesting observation is that in this harder scenario, our Accumulator outperforms the fine-tuning baseline consistently in all situations in CIFAR-100. Figure 3 *bottom* shows the corresponding results for clients of different tiers on CIFAR-100 after training by different algorithms. The accumulator+adapter architecture performs most favourably. This again shows our Accumulator works for anytime inference in a harder and more practical scenario. We also provide communication efficency in Appendix B.

**Personalization.** Since clients' class and data distributions are typically different and diverse, adapting the globally trained model through per-client personalization is an interesting task for FL. Based on the pertained model of multi-tier federated learning as the most practical scenario, we can adapt the model to the local data of each client efficiently. Specifically, the multi-tier FL pretrained model is fine-tuned for ten epochs for each client. We use noisy data for personalization and the final test to simulate personal data distributions. For the corrupted CIFAR-100, we simply use CIFAR-C (Hendrycks and Dietterich, 2019). To corrupt SpeechCmdv2 dataset, we add 60% background

Table 4: Personalized performance on CIFAR-100-C and corrupted SpeechCmdv2 with multi-tier constraints after FL training on clean data (figures in bracket show relative improvement to performance before personalization).

| Update part for personalization | CIFAR-100-C | | | SpeechCmdV2 |
|---|---|---|---|---|
| | IID (1000.0) | Non-IID (1.0) | Non-IID (0.1) | Bkg. Noise (0.6) |
| Full Model | 34.30±1.85(+32.19) | 37.75±2.21(+36.36) | 26.94±3.14(+25.85) | 90.21(+8.59) |
| L.W. Linear +PA | 34.67±1.21(+32.10) | 37.55±1.79(+35.21) | 26.63±2.21(+24.46) | 80.82(+6.40) |
| PA Only | 34.06±1.01(+31.49) | 37.02±1.27(+34.68) | 25.96±1.58(+23.79) | 81.40(+6.98) |
| L.W. MLP + PA | 31.69±1.42(+29.46) | 36.95±2.05(+34.80) | 22.30±2.96(+20.26) | 80.59(+6.42) |
| PA Only | 33.01±1.18(+31.73) | 37.38±1.34(+35.23) | 26.59±1.62(+24.55) | 80.82(+6.65) |
| Accumulator + PA | 38.23±1.72(+34.78) | 40.05±2.34(+36.90) | 31.26±2.98(+28.30) | **85.46**(+6.47) |
| PA Only | 37.33±1.37(+33.88) | 39.45±1.48(+36.30) | 30.24±1.69(+28.20) | 85.40(+6.41) |
| Client Token Only | **45.38**±0.96(+41.93) | **47.02**±1.13(+43.87) | **38.25**±1.30(+35.29) | 85.24(+6.25) |

Table 5: Left: parameter number of different methods. Right: Ablation study of Accumulator.

| Method | Parameter Num | | | Mean performance |
|---|---|---|---|---|
| Full Fine-tuning | 30.62M | | Full Accumulator | 48.55 |
| Parallel Adapter | 0.60M | | #self-attn blocks=3 | 48.86 (+0.31) |
| Layer-wise Linear | 0.46M | | No Replace | 45.81 (-2.74) |
| Layer-wise MLP | 8.95M | | No Residual | 43.34 (-5.21) |
| Accumulator | 3.17M | | No Parallel Adapter | 47.93 (-0.62) |
| Client Token | 0.38K | | Linear Head | 44.6 (-3.95) |

noise into each validation speaker and split them into two sets for personalization and the final test. From the results in Table 4, we can see that when tested on the corrupted data, the performances of all methods degraded. Now, our Accumulator shows outstanding performance among all competitors, including updating the full model during personalization. More interestingly, updating the Client token, which has an extremely small set of parameters, i.e. 0.38K parameters according to Table 5 (Left), in our Accumulator, works overall the best among all situations. And also, when comparing the test accuracy gains after personalization, we can see our Accumulator produces the most, especially when the Client token is solely used for personalization fine-tuning.

**Ablation Study.** In order to verify the contributions of different components proposed in our Accumulator, ablation studies were conducted based on the multi-tier based FL setup on CIFAR-100 ($\alpha = 0.1$), the hardest scenario. In Table 5 (right), the full accumulator indicates a transformer with depth=1, class token parallel connection, token replace mechanism, parallel adapter, and an MLP classification head. Among them, we find that parallel connection influences performance the most. This is mainly because DeiT's original class tokens are well learned on ImageNet, which are directly beneficial for model predictions in the downstream tasks. Removing token replacing reduces the effect of modulating the extracted features from DeiT. Thus the performance drops noticeably. And we also tried larger depth, i.e. 3, in the Accumulator, but it seemed not to produce much performance gain. Our Accumulator works complementarily with parallel adapters used but apparently does not rely on them to have excellent performance.

# 6 CONCLUSIONS

We explored the challenges and opportunities of leveraging pre-trained models for downstream federated learning tasks. While this might appear to raise communication costs and preclude diverse device capabilities from participating, we show that after introducing a novel parameter-efficient adapter module we can simultaneously capture the benefits of communication efficient adaptation, non-IID robustness, support for diverse device capabilities, and robust personalization.

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

Table 6: Transmitted message size (# communication round×# model parameters (M)) required to reach a target performance with multi-tier FL on CIFAR-100 and FEMNIST (best and second best).

| | Fine-tuning | L.W. Linear | L.W. Linear +Adapter | L.W. MLP | L.W. MLP +Adapter | Accumulator | Accumulator +Adapter |
|---|---|---|---|---|---|---|---|
| CIFAR-100 | 90×30.62 | 1500×0.46 | 140×1.06 | 610×8.95 | 165×9.55 | 40×3.17 | 100×3.77 |
| FEMNIST | 60×30.44 | 1500×0.28 | 160×0.88 | 1500×8.77 | 240×9.37 | 40×2.99 | 80×3.59 |

Table 7: The computational and memory budgets across all exit layers with Deit-S as the foundation model. The memory and compute footprint increases linearly with the layer at which the early exit is triggered. During training, the memory peak can be largely reduced if the base model is kept frozen, allowing in this way the participation of more constrained devices.

| Early Exit Layer | 0 | 1 | 2 | 3 | 4 | 5 | 6 | 7 | 8 | 9 | 10 | 11 |
|---|---|---|---|---|---|---|---|---|---|---|---|---|
| FLOPs(GB) | 0.21 | 0.40 | 0.57 | 0.67 | 0.95 | 1.13 | 1.31 | 1.50 | 1.69 | 1.88 | 2.07 | 2.67 |
| Params(MB) | 2.54 | 4.34 | 6.15 | 7.96 | 9.77 | 11.57 | 13.38 | 15.19 | 16.99 | 18.80 | 20.60 | 22.41 |
| Mem. Peak(MB) | 67 | 123 | 179 | 236 | 293 | 350 | 407 | 464 | 522 | 580 | 638 | 670 |
| Mem. Peak(MB)$_{frozen}$ | 47 | 62 | 98 | 134 | 170 | 206 | 242 | 279 | 315 | 352 | 389 | 426 |

# A  IMPLEMENTATION DETAILS

**Hyperparameter Settings**    The base setup is the same for all experiments, and we basically did not include any extra training tricks to be able to demonstrate that our proposed framework can be successfully adapted with **Anytime** and **Anywhere**. For all datasets, the initial learning rate for the SGD optimizer is set to 5e-3, and the batch size and training epoch on the client is set to 10 and 1, respectively. The `RandomResizedCrop` with scale=(0.05, 1.0) is used in training. The FedAvg McMahan et al. (2017) is the default aggregation algorithm.

**Conventional Federated Learning**    In this setting, the last layer of the foundation model is set as the only exit. The full foundation model can be used for training the additional parameters, *e.g.,* L.W. MLP, and Accumulator, *etc*. The baseline Fine-tuning indicates all parameters of the foundation model with an extra MLP head are free for tuning. The total training round is set to 500.

**Anytime Federated Learning**    The major difference from the conventional FL is that the exit layer $l \in [1, 12]$ for each client is randomly sampled from a uniform distribution for each training iteration. The total training round is set to 1500.

**Multi-tier based Federated Learning**    Exit layer is no longer produced by random sampling, but a fixed value that represents the current computing power of the client. In other words, each client will be assigned a permanent $l$ during the anytime FL. The distribution of clients' exit layer $l$ is balanced. The total training round is set to 1500.

**Personalization**    Following the data corruption methods in Hendrycks and Dietterich (2019), we apply a corruption policy with different severity $s \in [0, 5]$ as unique style for each client's local data. Then, we fine-tune the pre-trained multi-tier model with a predefined exit layer on these data for 10 local epochs.

# B  ADDITIONAL EXPERIMENTS

**Communication Cost of Different Methods.**    Table 6 shows the communication cost of different methods to reach a target test accuracy, which is set as the best performance of the L.W. Linear head here. We can see from the results that our Accumulator as such achieves the lowest cost to reach the target by only cost $40 * 3.17$ and $40 * 2.99$ message size for CIFAR-100 ($\alpha = 0.1$) and FEMNIST(Non-IID).

