# OpenReview forum: "Federated Learning for Inference at Anytime and Anywhere"
_ICLR.cc/2023/Conference — Submitted to ICLR 2023_

### Official Review · Reviewer_pRUk · 2022-10-20

**Confidence:** 3
**Correctness:** 2
**Technical Novelty And Significance:** 2
**Empirical Novelty And Significance:** 3
**Recommendation:** 6

**Clarity, Quality, Novelty And Reproducibility:**

* I think that the paper is not clear enough (see Strength And Weaknesses section).
* There is some novelty, mainly in the idea, yet there is no real technical novelty.
* It is not reproducible (see Strength And Weaknesses section).

**Strength And Weaknesses:**

Strength:
1. As far as I know, and admittedly it is quite hard to keep track of all the papers in FL recently, this paper is the first to suggest a method for using transformers in FL systems.
2. The solution is relatively simple.
3. The comparison to other approaches that also use transformers is comprehensive, although I think that the overall experimental section is not sufficient.

Weaknesses:
I found several problems with the paper and writing. I will group them into categories:
1. Writing:
a. It seems that the authors assume prior knowledge of concepts that are more common in the NLP/Vision-NLP communities (e.g., tokenizer, positional encoding, ViT architecture, CLS token, residual adapter). I believe that for many readers who engage in FL these concepts are not familiar (for example, FL is also very popular in the signal processing community). Hence, I think that an appropriate background should have been given in the appendix and the paper is incomplete without it.
b. The writing of the paper at times is very confusing. For example, already in the introduction, the authors present results with the accumulator, but it is not clear what it is and to what exactly it is compared until the method section. The authors refer to Fig. 2 on page 4 which in its description is talking about early exit, yet the explanation for what is early exit appears only on page 6.

2. Experiments:
a. Although, as I stated, the comparison within the realm of transformers is good, I think that the authors should try harder to compare to other methods as well. For example, using standard methods with a network that is pre-trained on some datasets (e.g., ImageNet). At the current state, it is hard to decouple the effect of the transformers, the pre-trained model, and the proposed solution.
b. The experiments with the corrupted CIFAR dataset are not clear to me. Which datasets are used and for what exactly? Also, I am aware of other studies that used corrupted CIFAR-100 in FL (see [1]), so it is not clear to me why not use these datasets and or compare to them if possible.
c. The authors show that their method has a low communication cost, which is great. I would like to understand better two other aspects of computationally which are also important in FL. How much memory does it take to train and make predictions with the proposed approach, and how much time does it take (wall-clock time) to make predictions on CPU and GPU? I wonder how it is compared to the standard FedAvg with simple architectures as commonly used.
d. Perhaps I missed it, but I didn't find an explanation on how did you search over the hyper-parameters of the model, or if a validation set was used (aside from the CIFAR-C experiments). In general, more details on the exact experimental section are missing. In the absence of code (which is a major flaw in the submission in my opinion) this information is important to assess the validity of the experiments.

3. Minor:
a. What is LDA partitioning? It is not mentioned in Hsu et al. (2019) paper as far as I can tell.
b. I think that Fig. 2 is not entirely clear.

[1] Achituve, I., Shamsian, A., Navon, A., Chechik, G., & Fetaya, E. (2021). Personalized Federated Learning with Gaussian Processes. Advances in Neural Information Processing Systems, 34, 8392-8406.

**Summary Of The Paper:**

In this paper the authors proposed to model clients in a federated learning (FL) setup as downstream tasks for large pre-trained models. The main component introduced by the authors is an accumulator module which is shared across the layers of a transformer and is learned based on a client token, CLS token and positional embeddings. To make the model suitable for clients with varying computational constraints the authors suggest to utilize exit points of the transformer. The authors evaluated their method against several natural baselines in this setting on common FL benchmarks.

**Summary Of The Review:**

Overall I think that using transformers in FL systems is a nice idea, but I think the paper has several flaws in the writing and experiments which makes it hard for me to recommend acceptance.

---

> ### Author Response · Authors · 2022-11-18
> **Response to Reviewer pRUk**
>
> **Q: 1. Writing.**
>
> A: We have now tuned the writing in the revised paper. And for the background details, we will include them in the camera ready if accepted.
>
> **Q: 2.a) Effect of the transformers, the pre-trained model, and the proposed solution.**
>
> A: Our claim is specific to the regime of pre-trained models and all reported results use the same pre-trained model for fairness. Our technical accumulator contribution is also specific to transformer architecture (and thus all our experiments were based on the same pre-trained transformer for fair comparison) so it’s not straightforward to disentangle our contribution from the transformer.
>
> Nevertheless, for completeness, we compare the vanilla Deit-S and ResNet-50 to show the effect of the different architecture design of the foundation model. We can see that pretrained Deit-S performs better, especially when data heterogeneity is more severe, the same as the observation in [2]. However, our accumulator, as a new integrated solution for FL, depends on the ViT arch, so it’s hard to decouple the effect of the transformer and our proposed method.
>
> | Pretrained Model | CIFAR-100 (alpha=100.0) | CIFAR-100 (alpha=1.0) | CIFAR-100 (alpha=0.1) |
> |:----------------:|:-----------------------:|:---------------------:|:---------------------:|
> |      Deit-S      |85.05|84.91|84.21|
> |     ResNet50     |76.34|75.82|74.69|
>
> **Q: 2.b) CIFAR-100-C exp.**
>
> A: We use CIFAR-C to evaluate a different example of personalisation beyond the typical class-skew typically used for FL. CIFAR-C contains different data distributions corresponding to different image corruptions (E.g., blur, etc). In our experiments, after training on regular CIFAR-100 for centralised FL, we conduct personalisation by assigning one CIFAR-C corruption/distribution to each device and personalising that device to the local data distribution. Please refer to the Personalization paragraph in Section 5.3 and Appendix A in the revised paper for detailed setup.
>
> This differs from [1], who just use CIFAR-C to test whether a FL model trained on clean data is robust enough to generalise to the CIFAR-C corruptions (but without any local adaptation to the specific corruptions).
>
> To demonstrate the stability of our method to noise, we also report results using the same testing setup as [1]: Train our model on clean CIFAR, and then test it on  CIFAR-C.
>
> |     Methods     | alpha=0.1 | alpha=1.0 | alpha=1000.0 |
> |:---------------:|:---------:|:---------:|:------------:|
> | Full FT |   72.35   |   72.89   |     73.24    |
> |  L.W. Linear   |   57.33   |   57.80   |     58.41    |
> |  L.W. MLP    |   58.12   |   58.44   |     59.03    |
> |  Accum   |   72.25   |   72.70   |     73.38    |
>
> **Q: 2.c) Memory and inference time.**
>
> A: The memory usage during training and inference with conventional FL is shown below. Please note that our method is not a competitor to FedAvg – we use FedAvg as the learning algorithm for all competitors in our paper. Our contribution is an architectural one for anytime prediction, so the competitor is the baseline architecture for anytime, DeiT-S [2] equipped with layer-wise heads and the various other alternatives we consider. Notably, Our Accum has similar memory consumption as a single MLP head. Due to using a Foundation Model, the difference in the inference time between our method and baselines under the conventional FL setting is not significant (about 1338.24 ms on RTX 2080 Ti and 3453.42 ms on Intel(R) Xeon(R) Gold 5118 CPU @ 2.30GHz). The results are measured by using the Event.elapsed_time in pytorch. However, for the communication cost to reach a target performance, the more important factor in FL, our Accum is significantly smaller than other baselines as shown in Table 6 in the Appendix.
>
> | Methods (bs=10)  | Train Memory Usage | Inference Memory Usage |
> |:--------------:|:------------------:|:----------------------:|
> | Deit-S |1.68G| 0.27G|
> |  ResNet-50  |2.08G|0.30G|
> |  MLP head  |0.46G|0.24G|
> |  Accum  |0.52G|0.25G|
>
> About the cost to make inferences, we emphasise that our Accum facilitates early exits (unlike regular DeIT/ResNet [2]) which means that training and inference cost is scalable in both time and memory (see Table 7 in the appendix and Fig 3) according to device capabilities, or even current device load. For example, standard DeiT-S architecture requires 2.67GFLOPS for inference, but ours can scale down 10x to 0.21GFLOPS.
>
> **Q: 2.d) Reproducibility.**
>
> A: We have now added the hyperparameter and validation details in the revised Appendix section A. Moreover, we add an anonymous link to the code in the abstract.
>
> **Q: 3. Minor.**
>
> A: We have now fixed the reference and tuned Fig. 2 in revision.
>
> [1] Achituve et al. Personalized FL with Gaussian Processes. NeurIPS 2021.
>
> [2] Qu et al. Rethinking Architecture Design for Tackling Data Heterogeneity in FL. In CVPR 2022.

---

> > ### Comment · Reviewer_pRUk · 2022-12-05
> > **Response to authors**
> >
> > I thank the authors for the response. As pointed out by Reviewer Fwvb and myself, the paper is not written well, and the changes made towards this rebuttal, to me at least, seem only cosmetic. However, I believe that the idea in this paper is interesting and important. I also think that the response of the authors to my concerns and to the concerns raised by other reviewers was good. Therefore, I decided to raise the score to 6. Please include the discussion here in the final version and give a proper background as promised in your response.

---

### Official Review · Reviewer_eyXk · 2022-10-25

**Confidence:** 3
**Correctness:** 3
**Technical Novelty And Significance:** 3
**Empirical Novelty And Significance:** 3
**Recommendation:** 5

**Clarity, Quality, Novelty And Reproducibility:**

The approach in this paper is well-introduced. The evaluation on the computation budget as well as the choice of adapter is also well-presented.

**Strength And Weaknesses:**

This paper raises an interesting topic in FL. However, this paper's presentation and evaluation do not provide enough support for its claims.
It would be better to provide justifications in the following domains.

1. Pre-trained models: in most of the practical FL settings, the client device cannot afford the memory to store the pre-trained models. How does the proposed approach generalize to FL from scratch?

2. Transformer models: this paper studies the transformer models for vision and acoustic models. Is it possible to provide an analysis of the language models, such as next-word prediction on the StackOverflow dataset?

3. Device diversity: How does the proposed approach support the diverse device capabilities? Should each client maintain the same pre-trained transformer models? If so, is the device capabilities diverse enough?


**Summary Of The Paper:**

This paper studies the federated learning problem. In particular, this paper aims to provide a unified framework for communication cost, robustness to data heterogeneity, and other challenges. To tackle these problems, this paper proposes an attention-based adapter module at each transformer block. In the federated optimization, only the lightweight adapter is trained. Empirical evaluation provides an analysis of the performance of heterogenous device capabilities, efficient personalization, and scalable-cost anytime inference.

**Summary Of The Review:**

In summary, this paper focuses on an interesting problem, and the evaluation in computation and communication is solid.

However, it would be better to have a better presentation of the paper with strong support for the claims.

---

> ### Author Response · Authors · 2022-11-18
> **Response to Reviewer eyXk**
>
> **Q: 1. Pre-trained models & generalisation to FL from scratch?**
>
> A: Pre-trained models are not necessarily huge models. In our current experiments we use DeiT-small which has only 22M parameters, less than ResNet50. Meanwhile other FL studies have already considered ResNet-110s [1] and even larger [3]. Our 22M parameter pre-trained DeiT is completely feasible for storing on recent mobile devices. It could of course be quantized if memory was an issue.
>
> More importantly, our technical contributions of early-exits allow devices with lower memory and/or FLOP constraints to run a fraction of the full DeiT model (Fig 3); and our parameter efficient accumulators mean that the actual comms cost of FL is much lower (see Fig 1, etc).
>
> We claim no contribution to training from scratch, only to the fine-tuning regime [2,3], where our technical contributions above improve on the vanilla fine-tuning baseline [2,3].
>
> **Q: 2. Analysis on StackOverflow?**
>
> A: Thanks for the suggestion. We could not finish the experiments for rebuttal due to the short timeframe. We strongly expect it will work because BERT is VIT-like, and we will include it in the camera ready if accepted.
>
> **Q: 3. Device diversity.**
>
> A: As our method supports early exits, different numbers of blocks of pre-trained transformers can be deployed to different devices according to their capacity. So they don't necessarily need to maintain the full sized pre-trained transformers. We now show the memory and flops computed for the different tiers of devices in Table 7 in the revised Appendix. We can see that the diversity (i.e. system heterogeneity) is well supported.
>
> [1] He et al, Group Knowledge Transfer, NeurIPS 2020.
>
> [2] Nguyen et al. Where to Begin? Exploring the Impact of Pre-Training and Initialization in Federated Learning. ArXiv:2206.15387.
>
> [3] Qu et al. Rethinking Architecture Design for Tackling Data Heterogeneity in Federated Learning. In CVPR 2022.

---

### Official Review · Reviewer_sDyc · 2022-10-25

**Confidence:** 4
**Clarity, Quality, Novelty And Reproducibility:** The paper is clearly written and novel.
**Correctness:** 3
**Technical Novelty And Significance:** 3
**Empirical Novelty And Significance:** 3
**Recommendation:** 6

**Strength And Weaknesses:**

The paper is quite well written in terms of clarify of the objective and applications. I also found the paper quite substantial in content as it tries to address both early exits (for inferencing)  as well as fine tuning of FMs in a single approach.

Here are some of the main concerns or ways in which the paper can be improved...

a) What is the unit of the compute budget in Fig. 3? In its absence it's difficult to gauge what kind of devices can be supported with various techniques.

b) Combining Table 1 and Table 5, it's not clear if the small improvement in accuracy is worth the additional number of parameters exchanged for the proposed approach as compared to the two other baselines. How do you justify that?

c) I am quite surprised to see that there is very little difference in accuracy in IID vs non-IID setting in all cases in Table 1. Normally we see significant drop in accuracy of final model when trained in a non-IID federated learning setting. Can you comment on that?

d) It would have been good to see some more baselines such as using a knowledge distillation based approach to train a smaller model from FM and aggregating that through FL. I wonder if you have any insights how such a baseline will perform.

**Summary Of The Paper:**

This paper presents an efficient way to fine tune foundation models in a federated learning setting. The authors propose a light-weight transformer adapter, called accumulator, weight of which are only trained during the federated learning as opposed to all the weights in the original foundation model. The authors claim that the propose method outperform other state-of-the-art methods for efficient fine tuning of FMs in a federated learning setting.

**Summary Of The Review:**

The paper provides a novel approach to take the problem of training large foundation model in a federated setting manner. With some more explanation on experimental results it should be a good paper for ICLR audience.

---

> ### Author Response · Authors · 2022-11-18
> **Response to Reviewer sDyc**
>
> **Q: a) The unit of the compute budget in Fig. 3.**
>
> A: The units of Fig 3 x-axis are the number of transformer layers executed before early exit. We now translate this into GFLOPS  in Table 7 in the revised Appendix.
>
> **Q: b) Small improvement v.s. additional parameters.**
>
> A: We assume the two other baselines the reviewer refers to are Linear Head & Linear Head+Adapter. Although the improvement margin is small in the conventional FL setting in Table 1, our method gets more noticeable improvements in other more challenging settings (C.F. Table 2-4). This justifies the small number of additional parameters.
>
> **Q: c) Close performance in IID vs non-IID settings in Table 1.**
>
> A: We treat this as a complement rather than a defect. The reason is that we use a pre-trained Transformer, which increases robustness to data heterogeneity as discussed in [1]. So the performance of Non-IID gets much closer to IID compared with conventional FL methods that train from scratch. Achieving this outcome, while simultaneously reducing the comms cost compared and supporting device heterogeneity – both through our accumulator –  is one of the major contributions of our study!
>
> **Q: d) Knowledge distillation-based baseline.**
>
> A: We use MobileViT-XS (2.3M) with about the same number of parameters as our Accum as students for knowledge distillation (KD) with teacher FM models Deit-S and Deit-B. From the results below in the conventional FL setting, we can see that our Accum outperforms the KD baselines significantly. In Particular, we can see KD baselines are sensitive to data heterogeneity without leveraging features of the foundation model. To train the student model from scratch, we employed an extra KL loss between teacher and student logits.
>
> | Teacher/FM | Student/Adapter | CIFAR-100 (alpha=100.0) | CIFAR-100(alpha=1.0) | CIFAR-100(alpha=0.1) |
> |:----------:|:---------------:|:-----------------------:|:--------------------:|:--------------------:|
> |   Deit-S   |   MobileViT-XS  |          72.38          |         70.52        |         68.09        |
> |   Deit-S   |   Accumulator   |          85.35          |         85.11        |         84.02        |
> |   Deit-B   |   MobileViT-XS  |          72.54          |         70.32        |         68.38        |
> |   Deit-B   |   Accumulator   |          87.36          |         87.08        |         86.21        |
>
>
> [1] Qu et al. Rethinking Architecture Design for Tackling Data Heterogeneity in Federated Learning. In CVPR 2022.

---

### Official Review · Reviewer_Fwvb · 2022-10-27

**Confidence:** 5
**Correctness:** 2
**Technical Novelty And Significance:** 2
**Empirical Novelty And Significance:** 2
**Recommendation:** 5

**Clarity, Quality, Novelty And Reproducibility:**

The paper’s major contribution is a lightweight framework for Transformer, and then is applied to a Federated setting. The paper is well written. There is no source code to evaluate the reproducibility.


**Strength And Weaknesses:**

Strength

1. This paper presents a technically sound method for efficient fine-tuning under a federated learning framework, which can reduce communication overheads and enable privacy preservation. Specifically, it regards a fixed pre-trained model as the feature extractor and uses a learnable aggregator (called Accumulator) with client indicators for deep representation learning.

2. Empowered by the neural structure, the proposed method supports "early exit", which can speed up inference and suit a variety of devices.

3. Compared with adequate baselines and competitors, the proposed method shows the best performance across several datasets.

Weakness

1. My biggest concern is the novelty of the proposed neural architecture. Essentially, regardless of the framework of federated learning, this work aims to introduce a new adapter for efficient fine-tuning. However, such an accumulator-style adapter can be seen as a simplified version of the Ladder Side-Tuning by  [1]. The simplification is done by only considering sequence-level features.

2. This paper claims the proposed method can alleviate severe data heterogeneity to achieve the purpose of "anywhere", but I did not see any specific neural design or learning method to target the data heterogeneity issue.

3. The paper is not well-written, making it not easy to follow and understand. For example, the passage under the title of Section 4 said that "our attention-based adapter Accumulator", but "Adapter" and "Accumulator" are regarded as different modules in the experiments. Meantime, please fix writing typos, e.g., "Competiters". The symbols in Eq 3 should be formally defined, such as M and alpha.

4. The definition of personalization is not accurate. Personalization in the federated setting is learning many personalized models simultaneously rather than learning only one model for a client. Therefore, the definition of Eq (2) and (9) are not typical federated personalization objective functions.

5. The experiment cannot support its claims. For example, in Table 1, the performance on FEMINST dataset is relatively too low compared to other work in the conventional FL settings.


Some recent papers [2,3,4] could be discussed to enhance the related work section by introducing federated learning on pre-trained models. It is worth noting that because the target topic is very new while the related papers are published recently, thus, the missing of these references won't impact the rating of this submission.

REFERENCE:

[1] Sung et al., LST: Ladder Side-Tuning for Parameter and Memory Efficient Transfer Learning.

[2] Yuanyishu Tian, et al., When Federated Learning Meets Pre-training, ACM TIST 2022

[3] Yue Tan, et al., Federated Learning from Pre-Trained Models: A Contrastive Learning Approach, arXiv 2022

[4] Orion Weller, et al., Pretrained Models for Multilingual Federated Learning, NAACL 2022


**Summary Of The Paper:**

The paper proposes to exploit pre-trained Transformer models in federated settings. Specifically, a novel attention-based adapter module is applied to collect information from each transformer layer/block, and then make an early prediction without needing all blocks.

**Summary Of The Review:**

The paper proposes a lightweight adapter to be combined with a pre-trained Transformer. This design can not only to be applied to solve the NLP downstream tasks and also to be aligned to the Federated task. However, the learning objective of personalization is not clearly defined to be matched to the federated setting.

---

> ### Author Response · Authors · 2022-11-18
> **Response to Reviewer Fwvb**
>
> **Q: 1. Comparison to “LST”.**
>
> A: LST and our Accumulator architecture are very different despite the superficial similarity at the abstract diagrammatic level, and they were proposed in completely different tasks and contexts.
> We summarise the differences as follows.
>
> - The philosophies behind are very different: LST doesn’t backpropagate through the foundation model (FM), and does not impact the FM’s feature extraction. Instead it only takes the FM as a fixed feature extractor. In contrast, our accumulator feeds into the FM, modulating the feature extraction process in the forward pass, and back-propagates through the feature extractor in the backward pass. This is a much stronger form of FM adaptation than the paradigm of LST which trains the side network simply using the FM as a fixed feature extractor.
>
> - Accumulator $\neq$ simplification of LST: as mentioned by LST (Sec 3.2), LST is “a lightweight version of the backbone transformer”, which means LST has the same number of layers as the FM, while Accumulator is recurrent, with a single module shared by all backbone layers, which serves as a memory efficient “skip” connection to the FM.
>
> - LST is not compatible with memory-efficient anytime inference: A key main reason why Accumulator has significantly lower communication cost (Figure 1) compared with other adaptor based or parameter-efficient transfer learning methods is that its representation is unified across layers thanks to the recurrent accumulator, which means that we don’t need to associate each early exit a task-specific linear head. An adaptation of LST to anytime inference would incur large memory cost due to the need to associate a different linear head with each layer.
>
> - LST would imply prohibitive comms cost in FL: (2) and (3) together mean that if LST was adapted to an FL context, the comms cost of LST + EE head parameters alone would be > 10x that of our total comms cost.
>
> - Finally, we remark that ICLR reviewer guidelines https://iclr.cc/Conferences/2023/ReviewerGuide dictate that material not published at a peer reviewed venue before May 28 2022, need not be compared. LST was neither published nor even uploaded to arXiv at this date, and is therefore definitely inadmissible as prior art.
>
> **Q: 2. “Anywhere” FL.**
>
> A: Our claim of improving “anywhere” (data heterogeneity) in FL is achieved in three ways: (1) using a pre-trained Transformer, which was shown in [1] to be robust to data heterogeneity (this part is not novel, and corresponds to our fine-tuning baseline) but (2) further improving on this to make it more communication and compute-efficient via our Accumulator architecture (as summarised in Fig 1, Fig 3, etc). Finally (3) Our accumulator architecture provides a client token which allows rapid and reliable personalisation to the unique heterogeneous data distribution of each client (Tab 4).
>
>
> **Q: 3. Writing.**
>
> A: Sorry for the confusion. Our Accumulator is a new type of adapter, which is complementary to the existing adapters. We will update the paper to disambiguate.  We have now fixed the typos and symbol definition issues. C.F. blue highlights.
>
>
> **Q: 4. Definition of personalization.**
>
> A: Our definition is correct. We have tuned the explanation around Eq.(2) to make it clearer. Eq 1 describes learning a single shared global model w^* for all clients; Eq 2 refers to each client independently fine-tuning that model to obtain a personalised model on their local data. This is the same pipeline as described in [2,3]. Each client conducts Eq 2 independently (but starting from the common global model w^*). Thus, we learn many personalised models simultaneously.
>
>
> **Q: 5. FEMNIST results.**
>
> A: Apologies for the conclusion. The explanation is that we used a partial set (10%) of the full FEMNIST with only 381 clients in our experiments (as mentioned in Sec 5.1), while previous work used the full set of 3550 clients. When we use the full set with 3550 clients, we can observe that our Accumulator can reach ~86% accuracy in the conventional non-IID FL setting, which outperforms the SOTA [4].
>
>
> [1] Qu et al. Rethinking Architecture Design for Tackling Data Heterogeneity in Federated Learning. In CVPR 2022.
>
> [2] Li, Tian, et al. Ditto: Fair and robust federated learning through personalization. In ICML 2021.
>
> [3] Oh J, Kim S, Yun S Y. Fedbabu: Towards enhanced representation for federated image classification. In ICLR 2022.
>
> [4] Prasad et al. Reconciling security and communication efficiency in federated learning. 2022. https://arxiv.org/pdf/2207.12779v1.pdf

---

> > ### Comment · Reviewer_Fwvb · 2022-12-12
> > **I will increase the rating.**
> >
> > Dear authors,
> >
> > Thanks for the rebuttal. It addresses part of my concerns. I will increase the rating accordingly. However, I am still not fully convinced by the rebuttal on Anywhere and the experiment results.

---

### Author Response · Authors · 2022-11-18
**To AC and all reviewers.**

We sincerely thank all reviewers for their thoughtful and constructive reviews of our manuscript. We provide responses to address the concerns of each reviewer point by point and upload a revised manuscript. The changes made in the revision are highlighted in blue.

In addition, we add our anonymous code for reproducibility in the following link:
https://anonymous.4open.science/r/Federated-Learning-for-Inference-at-Anytime-and-Anywhere-356A/README.md

Please let us know if you have any further questions. Thanks again for all your efforts!

---

### Decision · Program_Chairs · 2023-01-20

**Decision:**

Reject

**Justification For Why Not Higher Score:**

Reasons discussed in the meta-review.

**Justification For Why Not Lower Score:**

N/A

**Metareview: Summary, Strengths And Weaknesses:**

This paper presents a federated learning algorithm in the setting where each client is learning a transformer model. With the goal of efficient training, the paper presents a method which takes pre-trained transformers and adds an attention-based adapter module at each transformer block. The paper also attempts at learning a personalized model at each client.

The paper received mixed reviews initially. Some of the key concerns raised by the reviewers (some of which are shared by me too) were:

- The idea of using adapters in federated learning has been used in prior works.
- The idea of solving the data heterogeneity problem used in this paper doesn't involve much architectural novelty
- The personalization scheme used in the paper is rather basic, and the paper lacks clarity about how it is being done.
- The paper lacks clarity at various places.
- It also feels that the paper is trying to do too many things at once - handling heterogeneity, personalization, early exit - and each of these aspects individually is not looked at rigorously in the paper.

The authors' response was discussed. However, even after the discussion, some of the concerns still lingered and, in the end, no reviewer championed the paper strongly.

The paper is also missing some relevant recent works on federated learning and personalization using adapters, such as "Federated Learning with Partial Model Personalization" (Pillutla et al, ICML 2022)

Considering these aspects, the paper falls short of the acceptance criterion. The paper does have some interesting ideas such as early exit. However, in terms of overall contribution (technical novelty and experimental evaluation), it falls short. The authors are advised to incorporate the reviewers' feedback and more clearly place their work in comparison to the existing work on this topic.

The paper also has some other minor issues, such as

- The notation w_* in Eq 2 (personalization) is confusing. I suppose  w_*  refers to the global model which is then fine-tuned to get the optimal client-specific model. However, the way it is written, Eq 2 doesn't quite reflect that. In the rebuttal, the authors try to answer this (after Reviewer Fwvb raised this issue) but the lack of clarity still remains.
- The acronym "FM" is used at various places but hasn't been defined (I guess it means "foundation models" but should be defined).